# Microalgae Diversity in Interim Wet Storage of Spent Nuclear Fuel in Serpong, Indonesia

**DOI:** 10.3390/ijerph192215377

**Published:** 2022-11-21

**Authors:** Irawan Sugoro, Megga Ratnasari Pikoli, Dyah Sulistyani Rahayu, Marhaeni Joko Puspito, Syalwa Ersadiwi Shalsabilla, Firdaus Ramadhan, Diannisa Syahwa Rahma Fadila, Ade Cici, Devita Tetriana, Dinda Rama Haribowo, Mohammad Syamsul Rijal

**Affiliations:** 1Research Organization for Nuclear Energy, National Research and Innovation Agency, Jakarta 12400, Indonesia; 2Department of Biology, Faculty of Science and Technology, Universitas Islam Negeri Syarif Hidayatullah, Jakarta 15412, Indonesia; 3National Institute of Science and Technology, Jakarta 12630, Indonesia; 4Center for Integrated Laboratory, Universitas Islam Negeri Syarif Hidayatullah, Banten 15412, Indonesia

**Keywords:** *Chlorella*, diversity, microalgae, radioactive, spent fuel pool

## Abstract

The water quality in the interim wet storage of spent fuel (ISSF) needs to be monitored due to its function as a radiation shield. Water in ISSF pools must be free from microorganisms such as microalgae that live in a radioactive environment. Moreover, particular microalgae are capable of causing corrosion to stainless steel, which is a component of ISSF. Therefore, this study aims to determine the diversity of microalgae in the ISSF and those living in a radioactive environment, which cause corrosion. The microalgae were detected using the diversity and Palmer indices. The sampling of microalgae water was carried out by vertical filtration method at eight sites of ISSF. The results show that the diversity of microalgae (H′) was low due to radiation exposure in pool water, hence, only specific species can survive. The evenness (J′) of the microalgae was low, causing a high dominance index (C) value. Furthermore, the dominating species, namely, *Chlorella* sp. (*Chlorophyceae*), needs to be monitored because it has gamma radioresistance capabilities and can cause the corrosion of stainless steel.

## 1. Introduction

The interim wet storage of spent fuel (ISSF) in Serpong serves to receive and store spent fuel from the Multipurpose Reactor G.A. Siwabessy (RSG-GAS), and irradiated materials from radio metallurgy installation (RMI) and radioisotope installation (RI) [1]. This facility has a water pool in the middle with a size of 14 × 5 and a depth of 7.6 m. The walls, bottom of the pool, and storage racks for spent nuclear fuel are covered with stainless steel [2]. The facility is designed to accommodate 1448 spent nuclear fuel. Moreover, nuclear fuel is stored in a pool containing demineralized water produced using anion and cation exchange columns [3].

Demineralized water is used in the pool to function as a cooling system and radiation shield for spent fuel. The cooling system removes the decay heat generated by spent fuel through water circulation to maintain the temperature in the pool keep it under 35 °C [1]. A water quality that is not considered and does not pass the safety analysis can cause damage such as corrosion. Therefore, it releases the fission product radionuclides into the reactor cooling water and air in the environment [1,2]. The presence of microorganisms in spent nuclear fuel pools can cause corrosion of wall clad materials made of stainless steel [4]. This corrosion occurs on the walls, bottom of the pool, and spent fuel storage racks in the ISSF [2].

Several investigations into microorganisms in nuclear facility pool water have been carried out in different locations. Microorganism detection in ISSF has found corrosion-causing bacteria with an estimated population of 90,000 CFU/mL [3]. It was discovered that microbial blooms occur in the water pool of the nuclear facility in Sellafield, England, due to the algae *Haematococcus pluvialis* [5]. Furthermore, the microalgae diversity studies conducted in the radioactive environment of the R-9 industrial reservoir area, Lake Karachay, Russia, obtained a low result [6]. In mild steel materials, *Chlorella vulgaris* causes an fourfold increase in the average corrosion rate [7]. Another report showed that *C. vulgaris* to easily adhere to 316L stainless steel due to extracellular polymeric substances and cause corrosion after 21 days of incubation [8]. Corrosion due to microorganisms begins with the formation of biofilms, which can cause changes in the physicochemical properties of the steel surface [9,10]. Meanwhile, corrosion due to *C. vulgaris* is an electrochemical process with anodic and cathodic reactions, namely, the dissolution of FeO and reduction of oxygen [8]. *Chlorella vulgaris* attached to metal surfaces can form biofilms that cause corrosion due to the chloride ion content and oxygen produced by photosynthesis, which increases the concentration of dissolved oxygen (DO). Therefore, it accelerates the cathodic reaction and uptake of phosphate ions (P) for nutritional needs from *C. vulgaris* [7,8].

Research to detect the presence of microalgae in ISSF in Serpong has not been carried out. Therefore, this study aims to understand the microalgae ecology in ISSF pool water to prevent downtime and minimize corrosion caused by microorganisms in the facility. This work was carried out using the diversity and Palmer indices to provide information on microalgae in a radioactive environment and the potential to corrode stainless steel materials as a parameter of water suitability in nuclear facilities. Furthermore, the chemical and physical parameters of the water are measured as data to support research on microalgae life in nuclear facilities.

## 2. Materials and Methods

### 2.1. Sample Preparation

The ISSF facility is located in the Science and Technology Center, as part of the Radioactive Waste Technology Research Center, Serpong, Indonesia. Water sampling was carried out in December 2021 at 8 sites, where 3 are located in the pool (outlet, middle, and inlet) and 5 sites in the ISSF channel area based on the flow direction towards the main pool from RSG-GAS, PT. INUKI, and the channel junction, as presented in Figure 1. The microalgae sampling was carried out by filtering 20 L of water [11] vertically using a plankton net measuring 50 µm and a bucket size of 15 mL [12] to obtain a concentrated sample. The filtered microalgae sample was then transferred into a sample bottle and 10% Lugol iodine drops were added to preserve the sample [13].

### 2.2. Chemical–Physical Parameters of Water

Water chemical–physical parameters were measured in situ. Light intensity was measured using a digital lux meter LX 1010B, while the pH, temperature, total dissolved solid (TDS), and electrical conductivity (EC) were estimated with a multiparameter instrument, Hanna Instruments 9811-5 (Hanna Instruments Inc., Woonsocket, RI, USA). Subsequently, dissolved oxygen (DO) was evaluated with Hanna DO-5510, and water radioactivity using an STHF-R Water Proof High Dose Rate Probe by inserting the probe into the water. The maximum standard for water chemical–physical parameters for nuclear facilities refers to the book *Good Practice for Water Quality Management of Research Reactor and Spent Fuel in Nuclear Facilities* issued by the International Atomic Energy Agency (IAEA) world atomic agency [14].

### 2.3. Microalgae and Data Analysis

Samples were examined using a camera microscope (Olympus CX23, Olympus Corporation, Tokyo, Japan) with a hemocytometer at 400× magnification. The samples were observed on the hemocytometer’s entire surface due to the microalgae’s low density. The genus of microalgae was identified based on the morphology, the shape of the flagellum, the number of flagella, the flagellum direction, the chloroplast location, and the shape of the chloroplast. Reference sources used for identifying microalgae are van Vuuren et al. [15] and Bellinger and Sigee [12].

The identified microalgae genus was processed quantitatively ecologically. The abundance of microalgae was calculated using Lackey’s drop (microtransect) method [16], with the formula, $N=\frac{T}{L}\times \frac{P}{p}\times \frac{V}{v}\times \frac{1}{w}\;$, N = abundance of microalgae per liter (Ind/L), T = surface cross-sectional area of hemocytometer (mm^2^), L = area of one field of view (mm^2^), P = number of chopped plankton, p = number of fields observed, v = volume of sample in under the cover glass (mL), V = volume of filtered water sample with plankton net (mL), w = volume of filtered microalgae sample (L). The Shannon–Wiener diversity index (H′) was calculated by the formula, $H^{\prime }=-\overset{s}{\underset{i}{\sum }}Pi\ln\left(Pi\right)$, H′ = index of diversity, P = Ni/N, Ni = number of individuals of the first genus, N = total number of individuals of all genera, with the criteria H′ < 1 indicating low diversity; 1 < H′ < 3 indicating moderate diversity; H′ > 3 indicating high diversity [17].

The evenness of the microalgae was calculated using the Pielou evenness index (J′), with the formula, $J^{\prime }=\frac{H\prime }{H^{\prime }max}=\frac{H\prime }{\ln\left(S\right)}$, J′ = Pielou evenness index, H′ = Shannon diversity index, H′max = maximum possible value of H′ and it is equivalent to lnS, S = number of genera. The range for the evenness index is between 0–1. An index value close to 0 indicates low species evenness or that the number of individuals of each distribution type is very different. The index value is close to 1 indicates high evenness or that the spread of the number of individuals of each species is not much different or the same [18]. The microalgae dominance index was calculated using the formula, $C=\overset{s}{\underset{i=1}{\sum }}{\left(\frac{n_{i}}{N}\right)}^{2}$, C = dominance index, ni = number of individuals-i, N = total number of individuals. The dominance index is in the range of 0–1, with the provision that the smaller the value of the dominance index, fewer/no species dominate. The opposite applies where a large index value shows the dominance of a particular species [17].

The identified microalgae were calculated by estimating algae pollution index based on Palmer [18] to determine organic matter contamination in water samples. The genus and breed corresponding to the Palmer index will be scored. However, when it does not match the Palmer index, but is still found, a plus sign (+) is given. The minus sign (−) indicates that the microalgae were not found at the sampling sites. In this study, a pollution index assessment was carried out based on sample sites. The total score of the Palmer index is the sum of the scores of all the taxa found. Based on this value, the pollution category is known according to the numerical value formulated by Palmer. The numerical values are used to categorize the pollution index. When the value is 0–10, it is categorized as a lack of organic matter, 10–15 is moderate pollution, 15–20 is high pollution probability, and >20 categorizes high pollution.

Chemical–physical parameters as data supporting the life of microalgae were calculated using the water quality index (WQI) with the formula, ${\mathrm{WQI}}_{\mathrm{obj}}=\frac{\sum_{i=1}^{n}C_{i}P_{i}\;}{\sum_{i=1}^{n}P_{i}}$, where n = total number of parameters, Ci = value set for parameter i after normalization; Pi = relative weight of each parameter (1–4), as in Koçer and Sevgili [19]. The criteria for index scores are divided into 5 groups, namely, very good (91–100), good (71–90), moderate (51–70), bad (26–50), and very bad (0–25), based on Kannel et al. [20], to determine the water quality category in the spent fuel pool water.

## 3. Results

### 3.1. Community Structure of Microalgae and Water Quality in ISSF in Serpong

#### Composition and Abundance of Microalgae

The microalgae found in ISSF consist of 5 classes and 10 genera. The classes are *Chlorophyceae*, consisting of *Chlorella* sp., *Kirchneriella* sp., and *Chlamydomonas* sp.; *Cyanophyceae*, with *Chroococcus* sp., *Gloeocapsa* sp., and *Stephanodiscus* sp.; *Bacillariophyceae*, including *Nitzschia* sp. and *Navicula* sp.; *Zygnematophyceae*, with *Spirogyra* sp.; and *Trebouxiophyceae*, with *Dictyosphaerium* sp. *Chlorophyceae* and *Cyanophyceae* have the highest percentage in ISSF, about 30%, while *Bacillariophyceae* have a percentage of 20%, and *Zygnematophyceae* and *Treuboxiophyceae* of 10%, as shown in Table 1.

*Chlorella* sp. was consistently found at all sampling sites, followed by *Chroococcus* sp., which was present at seven sites. The total abundance of microalgae in the ISSF was 10,867 Ind/L. *Chlorella* sp. had the highest abundance, namely, 8916 Ind/L, followed by *Chroococcus* sp., with a value of 957 Ind/L (Table 1).

### 3.2. Shannon–Wiener (H′), Pielou (J′), and Simpsons Index (C)

The microalgae diversity index (H′) value has a value of 0.76, and is classified as low diversity. The evenness index value (J′) has a value of 0.31, indicating low evenness. Meanwhile, the Simpsons dominance index (C) has a value of 0.68, indicating the dominance of microalgae in nuclear facilities (Figure 2).

### 3.3. Water Quality and Palmer’s Pollution Index

The maximum limits for the parameters of light intensity, DO, TDS, and radioactivity are not regulated by the International Atomic Energy Agency, IAEA. Meanwhile, the institutions have regulated other parameters such as temperature, pH, and electrical conductivity (EC). This section shows the results of measuring the chemical–physical parameters of water in pools and channels from the ISSF and the maximum limits set by the International Atomic Energy Agency (IAEA), as presented in Table 2.

The light intensity measured at the location is different due to the installation of lights only in the pool area (K1–K5, K7), therefore, the channel area (K6 and K8) is 0, as shown in Table 2. Furthermore, the light exposure in the ISSF is not continuous, where the measured light intensity ranges from 0 to 40,000 lux, with the highest value at the K5 site and the lowest at K6 and K8. The water temperature in the channel and pool areas varies between 24.8–25.9 °C, below the IAEA maximum limit of <45 °C. In addition, this condition is also still suitable for microalgae. The pH value is within the range recommended by the IAEA, i.e., 4.5–7. The pH value that does not meet the specifications must be returned to the treatment area, and will be fed back into the system after meeting the required water quality [14].

The DO values measured were 3.9–5.5 ppm, with the highest value at K3 (pool outlet), and the TDS at all sites in the ISSF is 0 ppm. Since the conductivity value of the ISSF is 1.02–1.9 µS/cm, it does not exceed the recommended limit of <10 µS/cm [14]. The radioactivity in the water in the ISSF pool and channels was 2320.37 and 0.000024 mSv/h, respectively. The calculation results of the water quality index show good water quality in the ISSF (Table 2).

The total score range for all sites is between 0–10, indicating lack of organic matter pollution. Meanwhile, the total score for the algae pollution index based on genera is highest at site K8, and the lowest values are at sites K1, K2, K3, and K5. According to the Palmer, *Chlorella* sp., *Chlamydomonas* sp., *Navicula* sp., and *Nitzschia* sp. are pollution-resistant genera (Table 3).

## 4. Discussion

There are five classes of microalgae found in the ISSF facility, namely, *Chlorophyceae, Bacillariophyceae, Cyanophyceae, Zygnematophyceae*, and *Trebouxiophyceae*. These classes can trigger biofilm formation on stainless steel materials in nuclear facilities. *Chlorella* sp. can adhere to the material by releasing extracellular polymeric substances [7,8]. When the biofilm is not only formed by microalgae, but also other organisms such as sulfate-reducing bacteria, then nutrient exchange, gene transfer, signal transduction, and use of algae biomass by bacteria can occur [21,22]. Furthermore, uncontrolled water conditions in the facility can cause microbial bloom events, reduce visibility, slow deactivation, and disrupt fuel extraction operations [23,24].

*Chlorella* sp. from the class *Chlorophyceae* were almost always found in all sites, as presented in Table 1. This shows that *Chlorella* sp. is a microalgae that can survive in extreme environments. *Chroococcus* sp. is also found at almost all sites, including sites K6 and K8, where the light intensity value is 0 Lux. This is because the water in the ISSF is connected to the temporary fuel storage pool in the Multipurpose Reactor G.A. Siwabessy (RSG-GAS) and PT. INUKI. Moreover, microalgae can be sourced from both areas when the transfer of spent fuel takes place. When the transfer channel is closed, the water will only circulate in the channels and pools of the ISSF.

In this study, chemical–physical parameters were used as supporting data for the diversity of microalgae in nuclear facilities. The measurement results show that the water’s temperature, pH, and electrical conductivity (EC) are still below the maximum limit. Based on the life support for microalgae in water, all parameters are appropriate except for light intensity and radioactivity. The light intensity measured at sites K6 and K8 has a value of 0 Lux, and some genera are still found. The radioactivity parameter in this study is a limiting factor for microalgae life; therefore, only specific genera can live in this environment. According to the water quality index (WQI), the value obtained is in a good category, with a score of 75 (Table 2).

Generally, high light intensity increases the number of *Chlorophyceae* [25]. However, at sites K6 and K8, it was discovered that the measured light intensity value does not exist. Previous investigations have shown that members of the Chlorophyta grow under low light with mixotrophic and vertical migration abilities [26]. *Chlorella sorokiniana* can live in mixotrophic conditions and grow in autotrophic as well as heterotrophic conditions [27]. The large number of members of the *Chlorophyceae* group at site K8 may be caused by its ability to grow in mixotrophic conditions.

Chemical–physical parameters of water, such as DO, temperature, and pH, play an essential role in the distribution of these microorganisms [28]. Phytoplankton can live at temperatures of 20–30 °C [29] with an optimal temperature of 25 °C [30]. Cyanophyta and Chlorophyta showed good growth at high temperatures (20–30 °C), while Bacillariophyta had a less significant positive correlation temperature [31]. In this study, *Chlorella* sp. (Chlorophyta) and *Chroococcus* sp. (Cyanophyta) were the most abundant microalgae. Tolotti et al. [32] stated that temperature affects species specifically at the taxa level, causing variations in community composition. This is because of the variation of microalgae found in ISSF due to temperature.

The degree of acidity significantly affect the growth of autotrophic microalgae such as *Chlorella*, which grows at pH 4–10 [33]. The pH value of the water corresponds to the value measured at the facility to enhance the growth of *Chlorella* sp. It was also reported that the microalgae *Geitlerinema amphibium* (Cyanophyta) can grow and dominate in a radioactive environment with a pH of 8.4 [6]. Furthermore, DO will affect the photosynthesis process carried out by phytoplankton, increasing its levels and phytoplankton abundance [34,35].

The minimum DO level for the survival of aquatic organisms is 4 ppm [36]. The measured value in this study indicates the potential for the presence of microalgae to survive. In this study, the high DO content measured in the facility, primarily when produced from photosynthetic microalgae such as *Chlorella* sp., can trigger corrosion. There is a link between oxygen as the primary electron acceptor in accelerating the cathodic reaction, which will produce corrosion products [9]. The damage begins with the iron phosphate layer, which forms corrosion products in form of FeOOH and Fe_2_O_3_ [7].

The dissolved solids measured in the facility are 0 ppm, which indicates low organic matter in the channel and pool areas of the ISSF. When the TDS value is high, it shows the presence of organic matter that enters the water system. According to Rijaluddin et al. [37], an increased TDS value will affect the conductivity, while the low value of TDS and EC causes rapid light penetration into the water. Therefore, the process does not interfere with photosynthesis and abundance [38,39].

Water radioactivity is caused by piles of racks filled with spent nuclear fuel at the bottom of the pool [1]. Since the spent fuel in the pool still emits radiation, the level of radioactivity in the pool area is high. Meanwhile, the measured level of radioactivity is deficient in the channel area because there is no used material. This serves as a route for transferring spent fuel from the temporary spent fuel storage pool at the RSG-GAS and other facilities.

*Chlorella* sp. is almost always found at every site due to the radioactivity of water nuclear facilities. The effect of gamma irradiation on *Chlorella* sp. can cause a decrease in cell stability, size, chlorophyll content, and protein [40]. Pradhan et al. [41] stated that *Chlorella* sp. showed a radioresistance response to low doses of gamma radiation at 10, 50, as well as 75 Gy, and formed radioprotective phytochemicals that will protect cells from radiation damage. Moreover, gamma radioresistance occurs by changing physiological and metabolic processes in cells to form primary defense mechanisms in microalgae species [41,42]. Senthamilselvi [43] used *Chlorella* sp. to obtain mutants for increasing biomass and lipid production. This species had higher cell survival at doses of 100–600 Gy and lowered at doses of 800–900 Gy, and could also tolerate up to 900 Gy [44]. The effect of radioactive elements on microalgae discovered in this study needs to be further investigated to determine how it directly impacts cells and the communities formed, especially in nuclear facilities.

Radioactivity in the water is one of the limiting factors for microalgae life that can impact species richness and abundance in nuclear facilities. In this study, the total abundance of microalgae is 10,867 Ind/L, with *Chlorella* sp. as the genus with the highest abundance, namely, 8916 Ind/L, as shown in Table 1. Millan et al. [45] found a high abundance, dominated by *Planothidium frequentissimum* and *Crenotia angustio*, in water sources with radioactive content and morphological deformities. Shuryak [46] conducted an investigation on the species richness and abundance of plankton in radioactively contaminated waters, and observed their impact on structural as well as functional algal communities in the Dnieper River, Chernobyl, exclusion zone. Apart from the effect of radioactivity on abundance, dissolved oxygen can also have an effect. High dissolved oxygen content will increase the abundance of microalgae in the *Cyanophyceae* group [34].

The value of H′ in this study is less than 1, indicating that the diversity of microalgae is low, as shown in Figure 2, which illustrates the presence of dominant species in a location [47]. Low diversity can be caused by environmental conditions with a level of radioactivity; therefore, only certain microalgae can live. Similarly, Pryakhin et al. [6] conducted an investigation in the industrial reservoir area R-9, Lake Karachay, with a low microalgae diversity, and found four species, namely, *Geitlerinema amphibium* (Cyanophyta), *Scenedesmus quadricauda* (Chlorophyta), *Aulacoseira ambigua* (Bacillariophyta), and *Pleurochloris imitans* (Xantophyta). *Geitlerinema amphibium* predominates in this area. Duhovnaya et al. [48] stated that communities in extreme habitats have reduced species diversity, the reason being that sensitive species disappear, and this leads to less competition and high development rates of specific forms that are more resistant.

The evenness index value (J′) obtained is close to 0, which indicates low evenness or that the number of individuals of each genus has a very different distribution (Figure 2). When the index value is close to 1, it indicates an inevitable dominance of microalgae in each location [17]. In this study, the dominant microalgae are *Chlorella* sp. The increasing number of species that dominate can cause sensitive species to be lost due to pollution [49]. A high dominance index will impact the decreasing diversity and uneven distribution of other microalgae [50].

The low algal index value is in line with the WQI value. Based on this result, the low input of organic matter will affect the water quality. This indicates that the quality of water in ISSF is still below the required standards. The water quality inspection process in the cooling system was previously only carried out chemically and physically [51,52], while the microscopic examination was limited to detecting without identification [53]. Therefore, specific inspections can provide additional data and determine the appropriate means of control. This is important because microorganisms cannot be ignored, due to the potential for corrosion. The discovery of *Chlorella* sp. as pollution-resistant microalgae, based on the Palmer index, and its ability to cause the corrosion of stainless steel materials [8] is a warning to the authorities.

## 5. Conclusions

Low diversity (H′) of microalgae is caused by radioactivity in the pool, therefore, only certain species can survive. Based on this study, the dominating genus, *Chlorella* sp. (*Chlorophyceae*), needs to be monitored due to its gamma radioresistance capabilities, which causes corrosion in stainless steel. Apart from the existing chemical–physical parameters, the diversity and Palmer indices can be used for detecting microalgae, and as parameters of water suitability in nuclear facilities.

## Figures and Tables

**Figure 1 ijerph-19-15377-f001:**
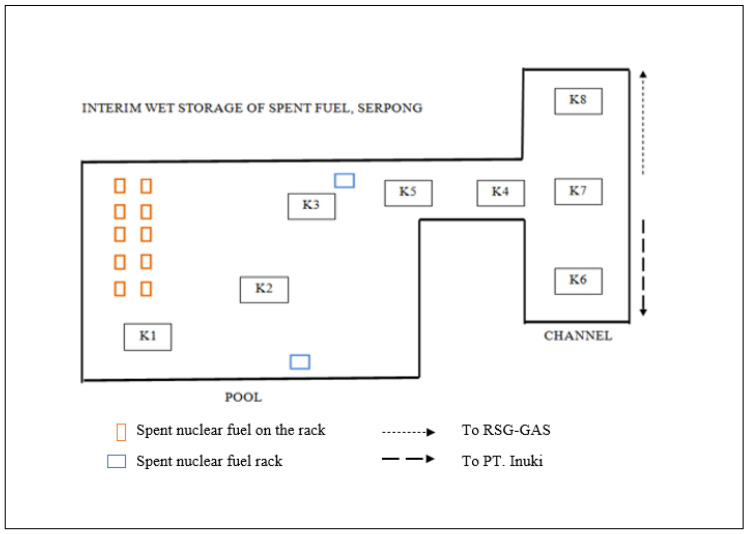
Sampling sites in pool and channels of ISSF (K1: inlet, K2: middle, K3: outlet, K4: before junction into the channel, K5: before junction into the pool, K6: left channel, K7: channel junction, K8: right channel).

**Figure 2 ijerph-19-15377-f002:**
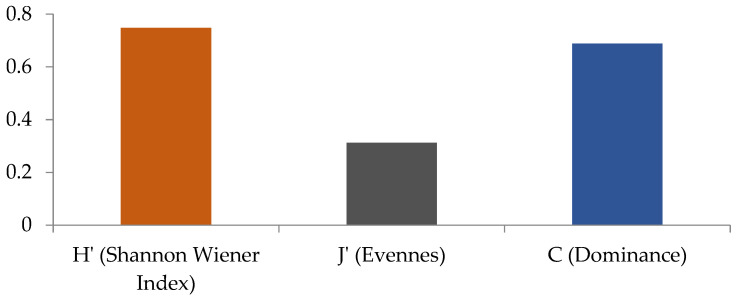
Shannon–Wiener (H′), evenness (J′), and dominance (C) values of microalgae in ISSF.

**Table 1 ijerph-19-15377-t001:** Microalgae in ISSF, Serpong (Ind/L).

Microalgae	K1	K2	K3	K4	K5	K6	K7	K8	Abundance (Ind/L)
***Chlorophyceae* (30%)**									
*Chlorella* sp.	✓	✓	✓	✓	✓	✓	✓	✓	8916
*Kirchneriella* sp.	-	-	-	-	-	-	-	✓	38
*Chlamydomonas* sp.	-	-	-	-	-	-	-	✓	115
***Cyanophyceae* (30%)**									
*Chroococcus* sp.	✓	✓	✓	✓	✓	✓	-	✓	957
*Gloeocapsa* sp.	-	-	✓	-	-	-	-	-	38
*Stephanodiscus* sp.	-	✓	-	-	✓	✓	-	-	191
***Bacillariophyceae* (20%)**									
*Nitzschia* sp.	-	-	-	✓	-	-	✓	-	77
*Navicula* sp.	-	-	-	-	-	✓	-	-	38
***Zygnematophyceae* (10%)**									
*Spirogyra* sp.	-	✓	-	-	✓	-	-	✓	344
***Trebouxiophyceae* (10%)**									
*Dictyosphaerium* sp.	-	✓	✓	-	-	✓	-	-	153
Total	2	5	4	3	4	5	2	5	10,867

Key: ✓ (present); - (not present).

**Table 2 ijerph-19-15377-t002:** Chemical–physical parameters of water in ISSF, Serpong.

Parameters	Sites								Avg.	Key	Limit of IAEA
	K1	K2	K3	K4	K5	K6	K7	K8			
Light Intensity (lux)	8000	6000	6000	32,000	40,000	0	22,000	0	14,250	NA	NA
pH	6	6.2	6.1	7	7	7	7	7	6.7	✓	4.5–7
T (°C)	25	24.8	24.9	25.9	25	25	25	25	25	✓	<45 °C
TDS (ppm)	0	0	0	0	0	0	0	0	0	NA	NA
DO (ppm)	5.2	5.3	5.5	4	4	3.9	4.5	4.4	4.6	NA	NA
EC (µS/cm)	1.02	1.02	1.02	1.9	1.9	1.9	1.9	1.9	1.9	1.57	<10 µS/cm
Water Radioactivity (mSv/h)	2320.37 *					0.000024 **			1160.18	NA	NA
WQI	75									NA	NA

Key: NA (not applicable), * (K1–K5), ** (K6–K8), ✓ (Required standard).

**Table 3 ijerph-19-15377-t003:** Pollution index of algal genera according to Palmer (1969) at 8 sites of ISSF.

Microalgae	K1	K2	K3	K4	K5	K6	K7	K8
** *Chlorophyceae* **								
*Chlorella* sp.	3	3	3	3	3	3	3	3
*Kirchneriella* sp.	−	−	−	−	−	−	−	+
*Chlamydomonas* sp.	−	−	−	−	−	−	−	4
** *Cyanophyceae* **								
*Chroococcus* sp.	+	+	+	+	+	+	−	+
*Gloeocapsa* sp.	−	−	+	−	−	−	−	−
*Stephanodiscus* sp.	−	+	−	−	+	+	−	−
** *Bacillariophyceae* **								
*Nitzschia* sp.	−	−	−	3	−	−	3	−
*Navicula* sp.	−	−	−	−	−	3	−	−
** *Zygnemathophyceae* **								
*Spirogyra* sp.	−	+	−	−	+	−	−	+
** *Trebouxiophyceae* **								
*Dictyosphaerium* sp.	−	+	+	−	−	+	−	−
Total score	3	3	3	6	3	6	6	7

Key: numerical values for pollution classification of Palmer are 0–10 (lack of organic pollution); 10–15 (moderate pollution); 15–20 (probable high organic pollution); 20 or more (confirmed high organic pollution); + (present); − (not present).

## Data Availability

Not applicable.

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
