# Peer review of "Microalgae Diversity in Interim Wet Storage of Spent Nuclear Fuel in Serpong, Indonesia"

_ijerph, 2022, doi:10.3390/ijerph192215377_

Round 1

Reviewer 1 Report

Generally the paper concerns the microalgae in the specific reservoir for a storage of  nuclear fuel rods . This topic is interesting, but the measurement methods are insufficiently described  namely all the measurements should be described shortly how to measure the microalgae, effective dose ect... what intruments was used.

Lines 73-74: „The pool  water was taken as much as 20 liters [10] and filtered to obtain a volume of 15 ml” What the authors filtred 20 liters to obtain 15 ml ?

Line 82: how the authors measured radioactivity?

In table 2 the IAEA maximum limits are missing.

Line 144 S/cm or mS/cm?

The pollution indexes defined by Palmer at the 1969 could be out of date? May be there is newer reference.

In my opinion the paper can be published after  giving short discription of the measurement method.

Author Response

Dear, 
We thank you for very helpful comments that have aided us to improve our manuscript.  Below we provide responses to each of the comments (Please see the attachment). We hope that the revised version will be suitable for IJERPH and we are most happy to make further improvements. 

Reviewer 2 Report

Authors should consider my comments to improve qualities of the manuscript.

1. There were quite a lot of grammatical errors and awkward sentences, please carefully revise the manuscript. Helps from professional editors are recommended.

2. I don't quite understand the sentence "the evenness of microalgae in the ISSF is also low and inversely proportional to the dominance" (in the abstract). Please revise the sentence

3. In introduction, please provide mechanisms of corrosion from microorganisms that are related to your work.

4. In introduction, please give some examples of other works that showed corrosion effects from microorganisms on spent fuel pool.

5. Give the name of country for "Lake Karachay".

6. Details for all measurement, including light intensity, pH, temperature, TDS, DO, EC, and radioactivity are needed.

7. In Results, how did you get the percentages for each class in ISSF?

8. Authors need to provide more definition and implication for all H', J', and C. How did you get these numbers?

9. What are the sensitivities for all chemical-physical parameters? For example, the light intensity, which was reported with very rough numbers.

10. Ns in Table 2 should be replaced with NA (Not Applicable).

11. How were the total scores and WQI calculated?

12. Why did only Chlorophyceae and Bacillariophyceae have pollution index, while others only had +/-?

12. Please provide reliable references to the claim that Cholorella sp. can survive in extreme environment?

13. The arrangement as well as details of the discussion were very confusing. Authors are recommended to carefully revise all discussion part.

14. Authors claimed that Chlorella Sp. showed a radioresistance response to low-dose gamma radiation. How about at high-dose case? 

15. Conclusions were poorly written. Please revise them.

Author Response

(The authors gave the same response as above.)

Reviewer 3 Report

This work presents "Microalgae Diversity in Interim Wet Storage of Spent Nuclear 2

Fuel, Serpong, Indonesia". This manuscript is recommended to be published after including and addressing the below listed comments with major corrections.

- The authors should eliminate the current grammatical and punctuation mark errors and also confirm the correct scientific English.
- The authors should write the complete terms of all abbreviations (including the instruments) before the first use in the abstract and main manuscript.
- The authors should clearly explain the innovation and importance of their work on the introduction of the manuscript. The introduction section needs to be elaborated.

Author Response

(The authors gave the same response as above.)

Round 2

Reviewer 2 Report

Authors have revised the manuscript well and answered all my comments. The manuscript is now acceptable for publication.